# An Optimal Design Method for Improving the Efficiency of Ultrasonic Wireless Power Transmission during Communication

**DOI:** 10.3390/s22030727

**Published:** 2022-01-18

**Authors:** Yu Li, Juan Cui, Gang Li, Lu Liu, Yongqiu Zheng, Junbin Zang, Chenyang Xue

**Affiliations:** Key Laboratory of Instrumentation Science & Dynamic Measurement Ministry of Education, North University of China, Taiyuan 030051, China; liyu950921@163.com (Y.L.); cuijuan@nuc.edu.cn (J.C.); lg965050@163.com (G.L.); liulu2235909526@163.com (L.L.); zhengyongqiu@nuc.edu.cn (Y.Z.); zangjunbin@163.com (J.Z.)

**Keywords:** ultrasonic wireless energy transmission, ultrasonic communication, impedance modulation, condition monitoring

## Abstract

Due to the excellent directivity, strong penetrability, and no electromagnetic shielding effect, ultrasonic waves have good potential for wireless energy transmission and information transfer inside and outside of sealed metal devices. However, traditional ultrasonic based energy transmission methods usually result in considerable energy consumption because of the impedance mismatch during the impedance modulation of the communication. This paper presents an optimal design method for efficient energy transfer during ultrasonic communication. The channel equivalent circuit model is established by only using the acoustic-electric channel scattering parameters. According to the equivalent circuit model, the channel impedance matches with a weak mismatch state is performed during the communication. In this way, the impedance modulation effect is ensured with a lower decrease in the energy transmission efficiency. Finally, the simultaneous energy transmission and impedance modulation are carried out through the 11 mm thick 304 stainless steel plate. The transmission power is 37.86 W with a transmission efficiency of 45.75%, and the modulation rate is 10 Kbps. Compared with the traditional methods, our proposed energy transmission efficiency is increased by 17.62%. The results verify the proposed method’s effectiveness and the high accuracy of the model. The proposed method has great engineering applications and broad prospects in condition monitoring of metallic environments.

## 1. Introduction

Modern equipment, such as engines, nuclear material containers, missiles, submarines, space stations [1,2,3,4], are protected by sealed metal structures to adapt to unique environments, such as high temperature and high pressure or special use requirements. In some cases, the equipment needs to penetrate the sealed metal casing to transmit necessary data and energy during long-term operation, for example, the health monitoring of aero engines and the wireless power and data transmission between some sealed compartments of submarines [5,6,7,8]. However, the sealed metal shell structure seriously hinders the development of the technology mentioned above, which is mainly expressed in the power supply of the internal monitoring system and the reliable return of monitoring data. Conventional technology uses perforation for power supply and data transmission. This will pose a higher challenge to the strength and sealing design of the structure. By contrast, ultrasound holds the characteristics of high energy density, good directionality, and no electromagnetic shielding effect. Moreover, the piezoelectric ceramics used to generate ultrasonic waves have similar acoustic impedance to metal [9,10]. Therefore, ultrasound for energy transmission and communication has broad application prospects in the internal state monitoring of sealed metal equipment. Connor [11] first put forward this idea of using ultrasonic waves to penetrate metal walls for simultaneously wireless energy transmission and data transmission. The idea uses 2ASK modulation based on the impedance modulation technique to achieve communication from inside to outside while transferring energy from outside to inside. This modulation method does not require a high-power carrier generation module for the internal system, and the key component for impedance modulation is the MOSFET, so the internal communication circuit is simple and has very low power consumption. Since then, researchers have carried out more in-depth and detailed studies. However, there are many challenges in the development of this technology. Such as, how to improve the energy transmission efficiency [12,13,14,15,16,17,18,19], how to increase the communication rate [20,21,22,23,24,25,26,27,28] and the integration and application of the system [29,30,31,32,33,34,35,36].

In terms of energy transfer, Hu [13] and Sherrit [17] established the acoustic-electric channel’s mathematical model and equivalent circuit models. The fundamental characteristics of the channel, such as frequency selectivity, voltage transmission gain, and energy transmission efficiency, are analyzed through the model. The numerical simulation results of their model showed consistency. In [18], the influence of different coupling modes between piezoelectric ceramics and metal walls on energy transmission efficiency is investigated. By comparing the three coupling methods of the mechanical clamp with grease, conductive epoxy joint, stress bolts, it is concluded that the conductive epoxy joint coupling method has better all-around performance in terms of practical applicability and transmission efficiency. Lawry [14,15] first applied the simultaneous conjugate impedance matching (SCIM) technology in this field. After the SCIM is performed, the channel transmission efficiency improves significantly, and 56.2 W of power is transmitted through the 9.53 mm thick HY-80 stainless steel plane. The channel energy transfer efficiency (the ratio of channel output AC power to channel input power) is 70.8%. The channel DC energy transfer efficiency (the ratio of the channel output DC power to the channel input power) is 19%. Yang [19] used a radio frequency AC-DC converter based on resonant rectifier bridge technology to convert the high-frequency AC of the ultrasonic output into a DC output, which can be directly used as the power supply of electronic equipment. The output power of the DC terminal is 15.7 W, and the channel DC energy transfer efficiency is 27.7%. Compared with previous studies, the system energy transmission efficiency is increased by 8.7%.

In ultrasound communication, Ashdown [20] proposed a simultaneous energy transmission and full-duplex communication scheme. It uses a frequency tracking algorithm to determine the working frequency suitable for energy transmission and communication. In the end, it can continuously power electronic devices with a total power of less than 100 mW and obtain a reliable communication rate of more than 30 Kbps. Zhang [28] proposed an ultrasonic communication system based on a single carrier frequency domain equalization, which has a lower peak-to-average power ratio while maintaining similar anti-multipath fading performance, compared with the traditional ultrasonic communication system The system prototype achieved an effective bit rate of 436 Kbps through a 70 mm thick steel barrier. Primerano et al. [21,24,25,27] is dedicated to the application of high-speed communication between the sealed compartments of ships. They use channel equalization technology and digital communication technology to achieve a communication rate of up to 30 Mbps.

In terms of system integration and applications, Lawry [6] designs and implements an ultrasonic energy transmission and communication system that can work at 260 °C for extreme environmental monitoring applications in some sealed metal cabins of ships. The system has a transmission power of 1 W and a communication rate of 50 Kbps. Chase [29] carried out practical application design for the metal pipeline application environment and implemented the system with FPGA, DSP, and MSP430 as microcontrollers. Finally, the handheld size of the entire system is realized. Charthad [30] has achieved the first proof of concept for a 4 mm × 7.8 mm sized implantable device that uses ultrasound for power transmission and successfully implemented end-to-end testing in Chickens. Rekhi [31] proposed the use of airborne ultrasound for wireless power transfer to mm-sized nodes, with intended application in the next generation of the Internet of Things (IoT). The experimental results show that 5 μW power can be transmitted when the distance is 1.05 m. In addition, the key technical specifications, such as aperture efficiency, dynamic range, and bias-free operation proposed in this study can effectively enhance the power output. For the structural health monitoring application of solid metal structures, Tseng [36] completely embeds the internal PZT and the internal electronic system into the metal and uses integrated circuit technology to reduce the size of the internal system to the millimeter level. Fu [33] proposes a demonstration application of using this technology to power the internal monitoring device of a nuclear waste container.

In general, many researchers have focused on ultrasonic-based wireless energy transmission and communication. However, the compatibility between energy transmission and communication in this process has not been noticed until now as far as we know. Traditional impedance modulation methods for the communication lead to a serious decrease in energy transmission efficiency. At the same time, there is no model to explain the impedance modulation process. Therefore, the innovation of this article is to establish an acoustic-electric channel equivalent circuit model based on the SCIM technique. Based on the model, we explain the impedance modulation in the communication process from the electrical point of view. In addition, an optimal design method to improve the energy transfer efficiency of the ultrasonic-based simultaneous energy transfer and communication system is proposed. Finally, we experimentally verify the accuracy of the model, the rationality of the impedance modulation interpretation and the effectiveness of the optimal design method. The optimal design method is simple to implement and can significantly improve the energy transmission efficiency of the channel without deteriorating the communication.

## 2. Design of the System

### 2.1. System Setup

Figure 1 shows a block diagram of the ultrasonic energy and information transmission system constructed in this article. The acoustic-electric channel consists of two piezoelectric discs and a 304 stainless steel plate. The two piezoelectric discs have the same resonant frequency and polarization direction, and both are in the thickness vibration mode. The piezoelectric sheet is coaxially pasted to both sides of the 304 stainless steel plate through an acoustic coupling agent. The stainless steel is electrically isolated from the test bench and the piezoelectric ceramic by a resin spacer and an acoustic coupling layer. Thus, the piezoelectric ceramic and the metal wall are connected floatingly. The specific information of the channel is provided in Table 1. The coupling agent we used is Araldite’s AralditeAV138M/HV998 hybrid glue. This coupling agent has high strength, good toughness and good resistance to environmental erosion. It is often used in ultrasonic technology products (such as high-power ultrasonic cleaning).

Due to the positive piezoelectric effect, the external piezoelectric ceramic (E-PZT) is driven by the alternating voltage to generate ultrasonic waves. The ultrasound penetrates the metal wall efficiently. The internal piezoelectric ceramic (I-PZT) receives transmitted ultrasonic waves. Owing to the inverse piezoelectric effect, the I-PZT outputs an alternating current signal of the corresponding frequency. The entire process realizes the conversion of two different energy forms of electricity and sound. The physical link involved in this process is called the acoustic-electrical channel. The output generated by the I-PZT is used as the internal electronic system power supply. When the internal sensor data needs to be transmitted back, the acoustic-electrical channel is also the data transmission channel. The ultrasonic wave is the carrier of data transmission. Ultrasound is partially reflected at the heterogeneous interface formed by the I-PZT and the metal wall. Changing the electrical impedance on the terminals of the I-PZT by the modulator, for example, opening the terminals or shorting the terminals, will change the acoustic impedance of the I-PZT. It will result in a change in the strength of the reflected echo from the heterogeneous interface. The E-PZT can sense the change in the strength of the reflected signal. The corresponding phenomenon is the change in the amplitude of the alternating voltage signal on the E-PZT terminals, and then the binary amplitude keying process is realized. The change in the amplitude of the alternating voltage on the E-PZT terminals has a one-to-one correspondence with the action of the internal circuit. Thus, it can be used for data return from inside to outside.

### 2.2. Simultaneous Conjugate Impedance Matching

The E-PZT and I-PZT are coaxially pasted on both sides of the metal wall through the epoxy, a strong coupling process. The input and output ports of the acoustic-electrical channel are interrelated and affect each other. Therefore, the design of the impedance matching network for the input and output ports cannot be performed alone. The SCIM technology enables the input and output ports to achieve perfect conjugate impedance matching simultaneously. This technology was proposed by Rahola [37] and was first applied by Lawry [14] in an ultrasonic wireless power transmission system penetrating metal walls. This technology designs the matching network for input and output ports by measuring the *S* parameters of the acoustic-electrical channel. Figure 2 shows the SCIM model of the channel. R_S_ is the internal resistance of the radio frequency power amplifier. While *R_L_* is the load impedance. The physical meaning of the *S* parameter is as follows:

*S*_11_: reflection coefficient of port 1 when port 2 is matched.*S*_22_: reflection coefficient of port 2 when port 1 is matched.*S*_12_: the reverse transmission coefficient from port 2 to port 1 when port 1 is matched.*S*_21_: forward transmission coefficient from port 1 to port 2 when port 2 is matched.

The ROHDE & SCHWARZ ZNB20 network analyzer(R&S, Munich, Germany) is utilized to obtain the *S* parameters of the acoustic-electrical channel. The input port and output port of the channel are respectively connected to port 1 and port 2 of the network analyzer. The parameter *S*_21_ represents the transmission coefficient from port 1 to port 2, so *S*_21_ reflects the energy transmission efficiency from the channel’s input port (port 1) to the output port (port 2). The channel energy transmission efficiency η is defined as:(1)$$\eta ={\left|S_{21}\right|}^{2}$$

The *Z* parameters of the channel can be calculated according to its *S* parameters. *Z*_0_ is the characteristic impedance of the network analyzer. The calculation expression [37] of the *Z* parameters is as follows:(2)$$\left\{\begin{array}{l} Z_{11}=Z_{0}\frac{(1+S_{11})(1-S_{22})+S_{12}S_{21}}{(1-S_{11})(1-S_{22})-S_{12}S_{21}}\\ Z_{21}=Z_{0}\frac{2S_{21}}{(1-S_{11})(1-S_{22})-S_{12}S_{21}}\\ Z_{21}=Z_{0}\frac{2S_{12}}{(1-S_{11})(1-S_{22})-S_{12}S_{21}}\\ Z_{11}=Z_{0}\frac{(1-S_{11})(1+S_{22})+S_{12}S_{21}}{(1-S_{11})(1-S_{22})-S_{12}S_{21}} \end{array}\right.$$

The impedance and required by the channel input and output ports to realize SCIM can be calculated by the following formula [37]:(3)ZS*=α1±Δ2Re{Z22},ZL*=α2±Δ2Re{Z11}
where
(4)α1=−2jRe{Z22}Im{Z11}+jIm{Z12Z21}α2=−2jRe{Z11}Im{Z22}+jIm{Z12Z21}Δ=(2Re{Z11}Re{Z22}−Re{Z12Z21})2−Z12Z212

Re{ } represents the real part of the *Z* parameter, and Im{ } represents the imaginary part of the *Z* parameters. In the calculation process, it is necessary to select the appropriate sign to ensure that the real parts of $Z_{S}^{*}$ and $Z_{L}^{*}$ are positive. After obtaining $Z_{S}^{*}$ and $Z_{L}^{*}$, the corresponding impedance matching network can be designed to make *R_S_* and *Z_S_* conjugate impedance matching and *R_L_* and *Z_L_* conjugate impedance matching to realize SCIM.

The *S* parameters of the matched channel are [37]:(5)$$S_{M}=F(Z-Z_{M}^{*}){\left(Z+Z_{M}^{*}\right)}^{-1}F^{-1}$$
where
(6)$$Z_{M}=\left[\begin{array}{cc} Z_{S}^{*} & 0\\ 0 & Z_{L}^{*} \end{array}\right]$$
(7)$$Z=\left[\begin{array}{cc} Z_{11} & Z_{12}\\ Z_{21} & Z_{22} \end{array}\right]$$
(8)F=2Re(ZS*)−1002Re(ZL*)−1

According to the previous definition, the transmission efficiency $\eta_{M}$ of the matched channel can be calculated by:(9)$$\eta_{M}={\left|S_{m21}\right|}^{2}\times 100\%$$

The *S_m_*_21_ is the forward transmission coefficient (*S_m_*_21_) of the scattering parameter matrix(*S_m_*) of the matched channel (including the matching network).

Figure 3a shows the maximum channel energy transfer efficiency at different operating frequencies when the channel is matched with conjugate impedance. It can be seen from the figure that the channel can achieve the maximum energy transfer efficiency of 68% at 1.051 MHz. The *S* parameters of the channel at this frequency are as follows:$$\left[\begin{array}{cc} S_{11} & S_{12}\\ S_{21} & S_{22} \end{array}\right]=\left[\begin{array}{cc} -4.58∠-158.49{}^{\circ} & -3.73∠-132.31{}^{\circ}\\ -3.78∠-132.61{}^{\circ} & -11.44∠112.64{}^{\circ} \end{array}\right]$$

The output impedance of the radio frequency power amplifier is usually 50 Ω. Thus, *Rs* is 50 Ω. The load impedance of I-PZT varies by the system it drives, but it is generally purely resistive. In this thesis, *R_L_* is selected as 100 Ω for matching network design. According to the paper, the matching network structure that realizes the impedance transformation from *Rs* to $Z_{S}^{*}$ and *R_L_* to $Z_{L}^{*}$ is not unique. In this paper, the input matching network and the output matching network both adopt parallel inductors and series capacitors structures for impedance transformation design. Figure 3b,c show the matching network that meets the SCIM conditions of the channel. Refer to the paper [19] for the specific design process. The nominal values of the capacitors and inductors used are marked in Figure 3c. It can be seen that the nominal values of the components used in the actual circuit do not agree with Figure 3b. However, the impedance matching circuits we built are verified by HIOKI’s IM 3536 LCR METER and can achieve the SCIM of the channel within the error tolerance. The main reasons for this difference are the distribution characteristics of the electronic component parameters and the presence of distributed inductance in the board’s alignment.

### 2.3. Impedance Modulation

Impedance modulation technology is the key to realizing internal data return in this research. The transducer and the metal wall have similar acoustic impedance. Furthermore, the two are coupled by an acoustic coupling agent. Thus, the E-PZT and the I-PZT are in a strong coupling state. The change of the load impedance on the I-PZT terminal will cause the input impedance (*R_IN_*) of the input port to change. The radio frequency power amplifier drives the E-PZT, and the internal resistance of the radio frequency power amplifier is Rs. Thus, this process can be equivalent to the voltage divider circuit model shown in Figure 4a. Figure 4b is a schematic diagram of the voltage across R_IN_ changing with time during impedance modulation. When the load *R_L_* of the internal system switches between *R_L_*_1_ and *R_L_*_2_ due to impedance modulation, the acoustic-electric channel input impedance *R_IN_* switches between *R_IN_*_1_ and *R_IN_*_2_ accordingly. It can be seen from the voltage division relationship that the amplitude of the AC voltage across *R_IN_* is also switched between the two values of Vmax and Vmin. Obviously, the change of the voltage amplitude of the *R_IN_* terminals corresponds to the state of the internal circuit of the channel, and the communication process from the inside to the outside can be realized by using the corresponding relationship. The above is the electrical explanation of the impedance modulation process.

The channel *Z* parameter matrix can be calculated from the channel *S* parameters. Which the channel *Z* parameters reflect the electrical characteristics of the channel. This paper establishes the channel equivalent circuit model based on the Z parameters. According to this model, the influence of the change of internal load *R_L_* on energy transmission and communication is analyzed. Figure 4c shows the channel equivalent circuit model established by the definition formula of the *Z* parameter matrix.

Based on the *Z* parameter definition Formula (10) and Kirchhoff’s Law, the input impedance *R_IN_* observed from the E-PZT at different loads *R_L_* is:(10)$$\left\{\begin{array}{l} {\dot{U}}_{1}=Z_{11}{\dot{I}}_{1}+Z_{12}{\dot{I}}_{2}\\ {\dot{U}}_{2}=Z_{21}{\dot{I}}_{1}+Z_{22}{\dot{I}}_{2} \end{array}\right.$$
(11)$$Z_{IN}=\frac{{\dot{U}}_{1}}{{\dot{I}}_{1}}=Z_{11}-\frac{Z_{12}Z_{21}}{R_{L}+Z_{22}}$$

Based on the electrical explanation of the impedance modulation process, the modulation coefficient *Ma* is defined to parameterize the impedance modulation effect. As shown in Figure 4b, *V*_max_ is the maximum envelope amplitude of the carrier, and Vmin is the minimum envelope amplitude of the carrier. The definition of *Ma* is:(12)$$Ma=\frac{\left|V_{\max}-V_{\min}\right|}{\left(V_{\max}+V_{\min}\right)/2}$$

The expression of *Ma* based on the channel parameters is:(13)$$Ma=\frac{\left|\left|Z_{IN1}\right|-\left|Z_{IN2}\right|\right|*2R_{S}}{\left|\left|Z_{IN1}\right|-\left|Z_{IN2}\right|\right|*R_{S}+2\left|Z_{IN1}\right|*\left|Z_{IN2}\right|}$$

Based on Equation (11), *Z_IN_*_1_ corresponds to *R_L_*_1_ and *Z_IN_*_2_ corresponds to *R_L_*_2_ in Equation (13). This means that the load on the channel changes from *R_L_*_1_ to *R_L_*_2_. Obviously, it shows 0 ≤ *Ma* ≤ 1. According to the definition, the greater the *Ma*, the more pronounced the carrier envelope level changes, the better the impedance modulation effect, the lower the design requirements for the subsequent envelope detection circuit, and the higher the communication reliability.

The Formula (13) shows that the key to impedance modulation is to change *R_L_* to change the channel input impedance. Figure 4d shows the two traditional impedance modulation methods. The modulation of the analog electronic switches *S* and *R_L_* in parallel is imaginatively referred to as downward impedance modulation. The modulation of the analog electronic switches *S* and *R_L_* in series is imaginatively referred to as upward impedance modulation. Under the two modulation methods of traditional impedance, the internal load impedance jumps between 0 and *R_L_* and between *R_L_* and infinity by switching on and off the analog switch. When communicating through the conventional impedance modulation method, the system has an energy transfer efficiency of 0 in half of the time (Assuming that the statistical probabilities of “0” and “1” in the bitstream are equal). The fundamental reason for this result is that the on and off the analog electronic switch caused a complete mismatch of the system impedance, leading to a severe drop in the system’s energy transmission efficiency.

Figure 4c shows the numerical simulation results of the influence of different *R_L_* on the input impedance and modulation coefficient of the acoustic-electrical channel when the channel is at the optimal transmission frequency $f_{M}$ = 1.051 MHz, and impedance matching is not performed (The initial load *R_L_* is 50 Ω).

It can be seen from the simulation results that the acoustic-electrical channel input impedance *R_IN_* decreases as the internal load *R_L_* increases, and the rate of decrease becomes slower and slower. When *R_L_* = 0 Ω, the input impedance is the maximum value of 28 Ω. When *R_L_* = 1000 Ω, the input impedance tends to 4.4 Ω. The channel modulation coefficient Ma presents a process of decreasing to zero at first and then increasing with the increase of *R_L_*. Since the initial equivalent impedance of the internal system is 50 Ω, when *R_L_* = 50 Ω, the channel input impedance does not change, that is, *Ma* = 0. When the traditional modulation method is adopted, the downward modulation method can obtain a better modulation effect than the upward modulation method, with *Ma* = 0.638 for upward modulation and *Ma* = 0.78 for downward modulation.

### 2.4. Trade-Off Design for Energy Transfer and Communication

The change of the load impedance is the key to realizing impedance modulation, but the impedance matching of the internal and external systems is also the key to energy transmission. In this part, this paper will analyze the effect of the variation of the internal load *R_L_* on the channel energy transfer efficiency and modulation coefficient in order to make a trade-off between reliable communication and efficient energy transfer. It is the main innovation of this thesis.

Under the action of voltage *U* and current *I* at the channel input port, the channel input power *P_IN_* is:(14)$$P_{IN}=\left|\frac{{\dot{U}}_{1}^{2}}{R_{IN}}\right|=\frac{{\left|{\dot{U}}_{1}\right|}^{2}}{\left|R_{IN}\right|}$$

The active power output by the channel output port is:(15)$$\begin{array}{l} P_{OUT}=\left|{\dot{U}}_{2}\right|*\left|{\dot{I}}_{2}\right|\\ =\frac{\left|R_{IN}Z_{22}-Z_{11}Z_{22}+Z_{12}Z_{21}\right|*\left|R_{IN}-Z_{11}\right|}{{\left|Z_{12}R_{IN}\right|}^{2}}*{\left|{\dot{U}}_{1}\right|}^{2} \end{array}$$

The channel energy transmission efficiency is:(16)$$\begin{array}{l} \eta =\frac{P_{OUT}}{P_{IN}}*100\%\\ =\frac{\left|R_{IN}Z_{22}-Z_{11}Z_{22}+Z_{12}Z_{21}\right|*\left|R_{IN}-Z_{11}\right|*\left|R_{IN}\right|}{{\left|Z_{12}R_{IN}\right|}^{2}}*100\% \end{array}$$

In practical applications, the more concerned is the relationship between the channel energy transmission efficiency and the modulation coefficient with the change of the load impedance after SCIM. In order to illustrate the modeling process, it is assumed that the channel’s input and output matching network adopt parallel inductance and series capacitance for impedance matching design. The equivalent circuit diagram of the channel after SCIM is shown in Figure 5a. *N*1 is the *Tx* matching network, *N*2 is the channel equivalent circuit, and *N*3 is the Rx matching network. The acoustic-electrical channel after SCIM can be regarded as a new two-port network *Nt* formed by cascading two-port networks N1, *N*2, and *N*3 in sequence. From the previous derivation process, it can be known that only the *Z* parameter matrix of the two-port network *Nt* is needed to obtain the relevant characteristics of the channel we want.

The *Z* parameter of the two-port network *Nt* can be calculated from the matching network parameter and the channel *Z* parameter. The specific calculation steps are as follows:Obtain the *Z* parameter matrix of the network *N*1, *N*2, and *N*3:

According to the definition of the *Z* parameter matrix of the two-port network and the channel S parameter, the *Z* parameter matrices of *N*1, *N*2, and *N*3 are as follows:(17)$$Z_{N1}=\left[\begin{array}{cc} Z_{1} & Z_{1}\\ Z_{1} & Z_{1}+Z_{2} \end{array}\right]$$
(18)$$Z_{N2}=\left[\begin{array}{cc} Z_{11} & Z_{12}\\ Z_{21} & Z_{22} \end{array}\right]$$
(19)$$Z_{N3}=\left[\begin{array}{cc} Z_{3}+Z_{4} & Z_{3}\\ Z_{3} & Z_{3} \end{array}\right]$$

2.Calculate the *T* parameter matrix of the network *N*1, *N*2, *N*3:

Table II shows the mutual conversion relationship between the *Z* parameter and the *T* parameter of the two-port network. The T parameter matrices corresponding to *N*1, *N*2, and *N*3 are *T*1, *T*2, and *T*3, respectively.

3.Obtain the channel T parameter matrix after SCIM:

Let *T_NX_* be the *T* parameters of the networks *Nt*. *N*1, *N*2, *N*3, and *T* parameters have the following relationships:(20)$$Z_{Nt}=T_{N1}T_{N2}T_{N3}$$

4.Obtain the *Z* parameter matrix of the network *Nt*

According to Table 2 [38] and the *T* parameters obtained in step 3, the *Z* parameter matrix of the network *Nt* can be easily obtained. Since the *Z* parameter matrix of the network *Nt* obtained by the above process has complex algebraic expressions, and the mathematical software MATLAB can easily implement the above numerical calculation process, the algebraic expressions of the *Z* parameter matrix of *Nt* are not explicitly listed in this paper.

Figure 5b is the numerical calculation result of channel energy transmission efficiency and modulation coefficient with the change of internal load impedance after channel SCIM. It can find that the modulation coefficient is not linear with the internal load impedance. Assuming that *Ma* = 0.2, it can meet the design requirements of reliable data return and envelope detection. If the downward impedance modulation method is used, only a 122 Ω resistor needs to be connected in parallel to reduce the equivalent load resistance of the internal system from 100 Ω to 55 Ω. Assuming that the statistical probabilities of data “0” and “1” in the bitstream are equal, the energy transfer efficiency of the system will be 68% for half of the time (the efficiency when the channel is fully impedance matched), and 59.1% for the other half of the time (the efficiency when impedance modulation leads to impedance mismatch), so the average energy transfer efficiency of the system is theoretically a transfer efficiency of 63.55%. When the upward impedance modulation method is used, only an 87 Ω resistor in series is required to increase the equivalent load of the internal system from 100 Ω to 187 Ω. The analysis and calculation process is the same as described above. The transmission efficiency of the channel after series resistance is 59.0%, and the expected value of system energy transmission efficiency during communication is 63.5%. In the case of the same modulation coefficient, the expected value of the energy transmission efficiency of the downward modulation method and the upward modulation method are approximately equal. However, the circuit of the downward modulation method is simple to implement, so it is more reasonable to adopt the downward modulation method, which is the optimization design method of communication and energy transmission proposed in this article. If the traditional downward modulation method is used, the expected energy transmission efficiency is 34%. With guaranteed communication quality, the method in this paper improves the expected value of energy transfer efficiency by 29.5% compared with the conventional downward modulation method.

## 3. Experimental Validation

### 3.1. Instruments and Methods for Experiments

A laboratory validation system was established to verify the correctness of the model and the feasibility of the optimization design method proposed in this paper, as shown in Figure 6. A GWINSTEK AFG-2225 function generator generates a continuous electrical signal at the desired frequency with an output impedance configuration of 50 Ω. The continuous ultrasonic electrical signal is driven by a homemade radio frequency power amplifier module with an output impedance of 50 Ω. A self-made amplifier with an output impedance of 50 Ω is used in our experiments. The power rating of the amplifier is 100 W, and the open-circuit output peak-to-peak voltage does not exceed 300 V. A GWINSTEK GDS-3504 oscilloscope is used to test the peak voltage *V_RL_* across the external transducer terminals at different internal loads *R_L_*. HIOKI’s IM 3536 LCR METER tests channel input impedance at different internal loads *R_L_*. Two types of internal loads are used: chip resistors (error 0.1%) and high power non-inductive resistors (error 1%). Chip resistors have high accuracy and a wide range of selectable resistance values, but cannot withstand high power. The high-power non-inductive resistor has the characteristics of bottom inductance and can withstand high power, but has fewer selectable resistance values. In the subsequent experiments, the chip resistors were used to test the modulation coefficients and the high-power non-inductive resistors were used for high-power transmission. ZLG’s PA5000H power analyzer is used to test the input power and output power of the channels, noted as P_IN_ and P_OUT_, respectively. They use the input and output port matching network designed in Section 2.

The matching network enables the channel to achieve impedance matching with *R_S_* = 50 Ω and *R_L_* = 100 Ω at the same time. Therefore, in the experiment, the system’s modulation coefficient can be calculated according to the expression (21), and the energy transmission efficiency can be calculated by the expression (22).
(21)$$Ma=\left|\frac{V_{RL}-V_{100}}{(V_{RL}+V_{100})/2}\right|$$
(22)$$\eta =\frac{P_{IN}}{P_{OUT}}\times 100\%$$
where *V_RL_* represents the amplitude of the voltage at the channel input port when the channel load resistance is *R_L_*. *V*_100_ represents the amplitude of the voltage at the channel input port when the channel load resistance is 100 Ω. *P_IN_* represents the channel input power and *P_OUT_* represents the channel output power.

### 3.2. Verification of the Channel Model

When SCIM is not performed, the input impedance and modulation coefficient of the channel are tested by varying the *R_L_* value from 0 to 500 Ω to verify the correctness of the model. The specific measurements configuration is shown in Figure 7a. When testing the input impedance of the channel under different loads, the function generator and oscilloscope are removed from the test system. On the contrary, the impedance analyzer is removed from the test system when testing the modulation coefficient

### 3.3. Application of the Optimal Method

After the channel realizes SCIM, experiments are designed to verify the feasibility of the optimized design method. Figure 8a shows the specific configuration of the experimental system. The rectifier bridge is composed of 4 STPS10L60D Schottky diodes. The input impedance of the rectifier bridge at the operating frequency measured by the impedance analyzer is 27.35 − 14.22i, so a power inductor with a nominal value of 2.2 μH is connected in series at the front end of the rectifier bridge to eliminate its capacitive reactance. The modulation resistance Z_L_ adopts a power non-inductive resistance with a nominal value of 100 Ω. An NTTFS4930NTWG MOSFET acts as an analog electronic switch. In the experiment, the channel modulation coefficient curve was tested. At this time, the gate of the MOSFET remains low, and the radio frequency power amplifier is removed. The signal generator directly drives the channel. In addition, when the channel is at the optimal operating frequency point, the energy transmission efficiency of communication and non-communication is tested respectively.

## 4. Discussion

Figure 7b shows the comparison between experimental and theoretical values of the input impedance and modulation coefficient when the channel load (*R_L_*) is varied between 0 and 500. The experimental and theoretical values of the input impedance curve are in good agreement. The experimental value of the later part of the modulation coefficient curve has a certain deviation from the theoretical value, but the error is within the allowable range. In the author’s opinion, the reason for this error phenomenon is as follows: as the load resistance increases, the channel input impedance gradually decreases and the rate of decrease becomes smaller and smaller. From the electrical explanation of impedance modulation in Section 2, the smaller the input impedance, the more obvious the measurement error of the modulation coefficient. In general, the correctness of the channel equivalent circuit model and the rationality of the electrical explanation of the impedance modulation process proposed in this paper have been verified by this experiment.

Figure 8b,c show the test results after channel SCIM. The experimentally measured modulation curve is consistent with the theoretical curve as a whole. When the channel is not communicating, there is a high resistance state between the source and drain of the MOSFET, so the modulator is not connected to the system. The experimentally measured channel output power is 51.53 W at an input power of 91.60 W. Therefore, the transmission efficiency of the channel is 56.25%. Although increasing the drive capability of the amplifier can enhance the power output of the channel, considering the nonlinear characteristics of the piezoelectric material, the mechanical properties of the coupling layer, and the electrical safety of the system, the higher the drive power is not better. If traditional modulation methods are used (Assuming that the statistical probabilities of “0” and “1” in the bitstream are equal), the theoretical value of the transmission efficiency during communication is 28.13%. when the gate of the MOSFET is driven at a rate of 10 Kbps and the binary voltage in the format of ‘01010101···’ (simulation of communication), the input power of the channel is 82.75 W and the output power is 37.86 W. Therefore, the energy transfer efficiency of the channel is 45.75%, which is 17.62% higher than the energy transfer efficiency based on the conventional modulation method. At this time, the modulation coefficient reached is 0.175. Although the test data in this experiment has a certain deviation from the theoretical value, it is completely acceptable when considering the following reasons: Firstly, the power amplifier used in this experiment is a self-made circuit module, and its output impedance cannot guarantee a constant 50 Ω in the entire working frequency band. Secondly, due to the distribution characteristics of electronic component parameters, it is difficult to realize an impedance matching network that is exactly the same as the design in practice. Thirdly, from the introduction of the experimental system, the modulator in this article is equivalent to a 127 Ω resistor in parallel across the load *R_L_*, which deviates from the design value of 122 Ω.

Figure 9 shows the carrier waveforms when the MOSFET is driven directly with the bitstream generated by the microcontroller. It can be seen that when the communication rate is 10 Kbps, the high and low variations of the modulated carrier envelope are evident and clear. Therefore, the channel is able to communicate reliably at this rate. This rate is comparable to other studies [20,24,26,35,39].

## 5. Conclusions

This paper establishes an equivalent circuit model that can effectively predict the characteristics of channel energy transmission and impedance modulation. Based on this model, an optimized design method for ultrasound based simultaneous energy transmission and communication is proposed. The feasibility of the optimized design method proposed in this paper is verified through experiments. In the experimental test, it can penetrate an 11 mm thick 304 stainless steel plate for simultaneous ultrasonic energy transmission and communication. Based on the optimization method, the transmission power during communication is 37.86 W, and the energy transmission efficiency is 45.75%, which is 17.62% higher than that of the traditional method. This optimized design method is simple to implement and can significantly improve the energy transmission efficiency of the channel. It has broad application prospects and immense engineering application value in not only wireless monitoring of metal environment but also other self-powered fields, such as underwater communication and implanted devices research. Future research will focus on system integration and high reliability design.

## Figures and Tables

**Figure 1 sensors-22-00727-f001:**
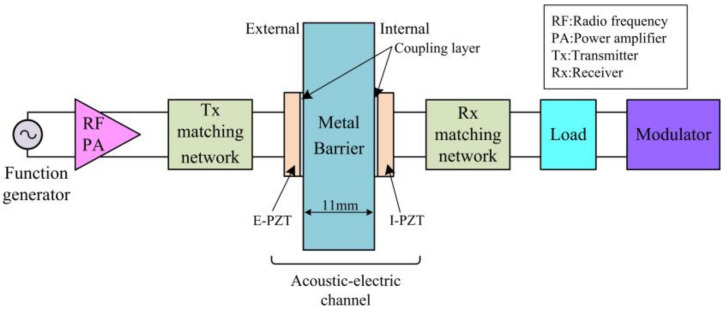
Block diagram of the ultrasound based simultaneous energy transmission and communication system.

**Figure 2 sensors-22-00727-f002:**
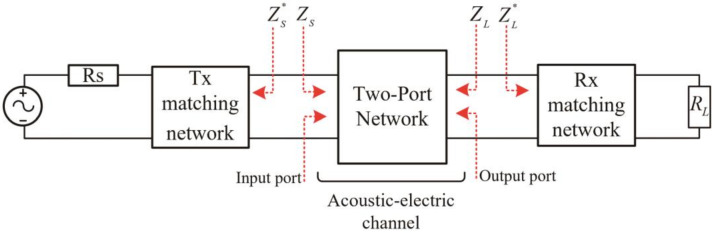
Simplified model of simultaneous conjugate impedance matching.

**Figure 3 sensors-22-00727-f003:**
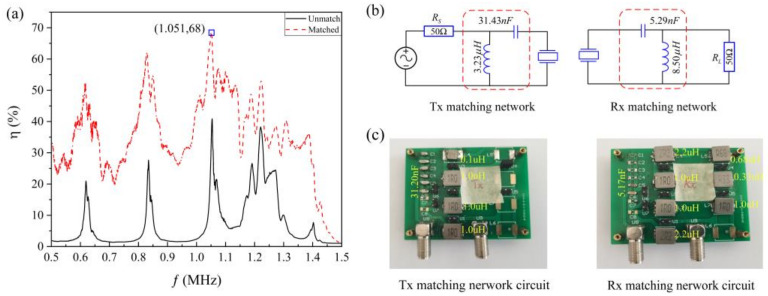
Simultaneous conjugate impedance matching network design. (**a**) Energy transfer efficiencies of matched and unmatched channels. (**b**) Matching network designed for SCIM. (**c**) Matching network built for SCIM.

**Figure 4 sensors-22-00727-f004:**
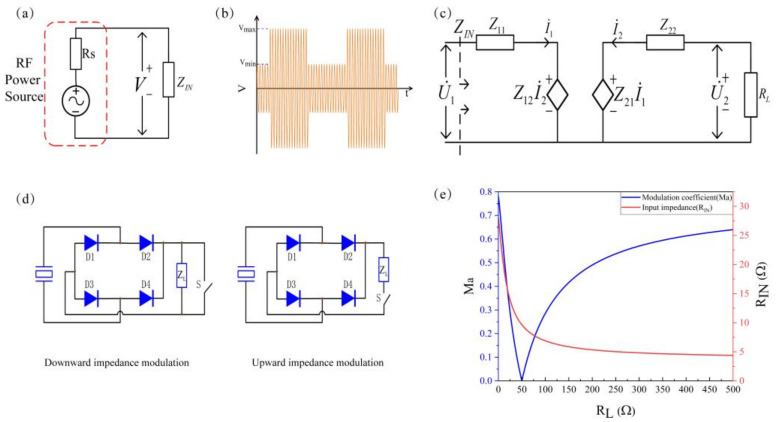
Impedance modulation. (**a**) Impedance modulation circuit model. (**b**) Modulated carrier on the terminals of E-PZT. (**c**) Equivalent circuit model of the channel. (**d**) Traditional impedance modulation method. (**e**) Modulation coefficient and Input impedance as the function of load resistance.

**Figure 5 sensors-22-00727-f005:**
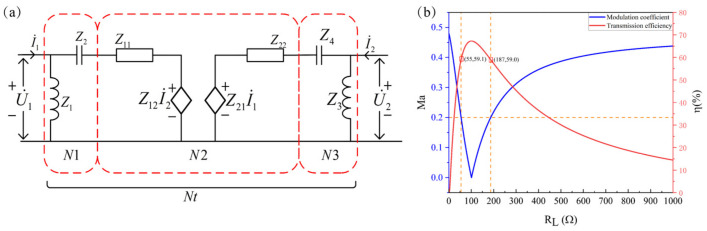
Trade-off design for energy transfer and reliable communication impedance modulation. (**a**) Matched channel equivalent circuit. (**b**) Modulation coefficient and transfer efficiency as the function of load resistance.

**Figure 6 sensors-22-00727-f006:**
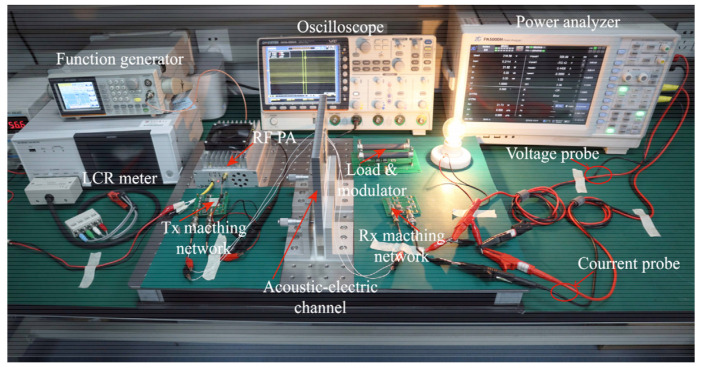
Experimental system (Easily light up a 20 watt incandescent lamp).

**Figure 7 sensors-22-00727-f007:**
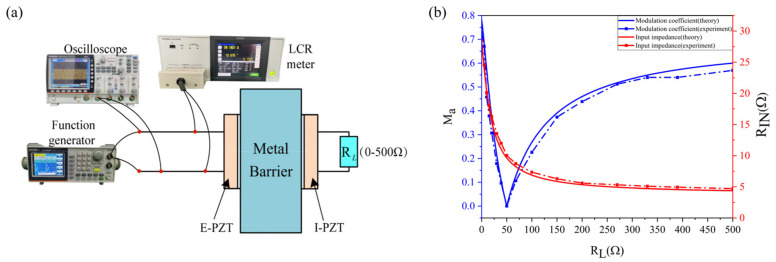
Verification of the Channel Model. (**a**) Measurement configuration for verifying the channel circuit model. (**b**) Comparison of experimental and theoretical values of input impedance and modulation coefficient before SCIM.

**Figure 8 sensors-22-00727-f008:**
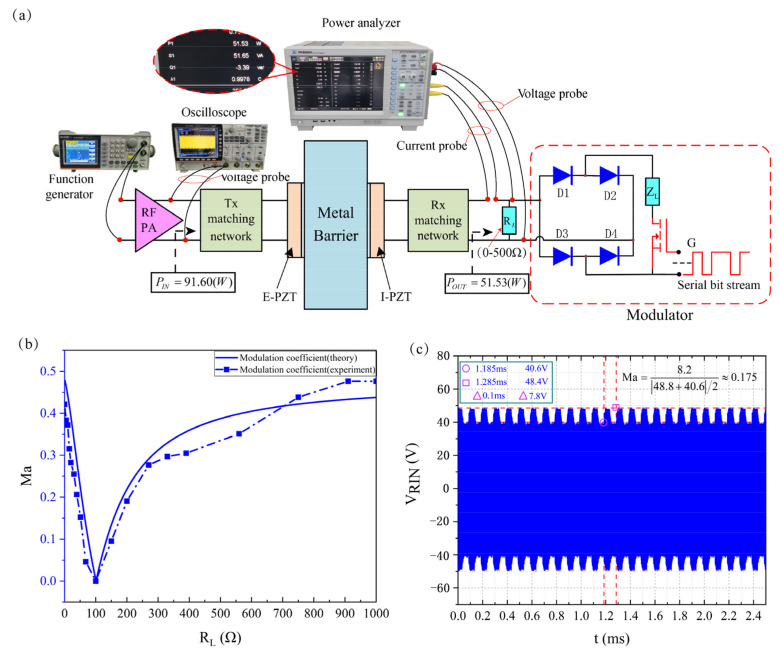
Application of the optimal method. (**a**) Measurement system configuration. (**b**) Comparison of experimental and theoretical values of modulation coefficient after SCIM. (**c**) Modulated carrier wave.

**Figure 9 sensors-22-00727-f009:**
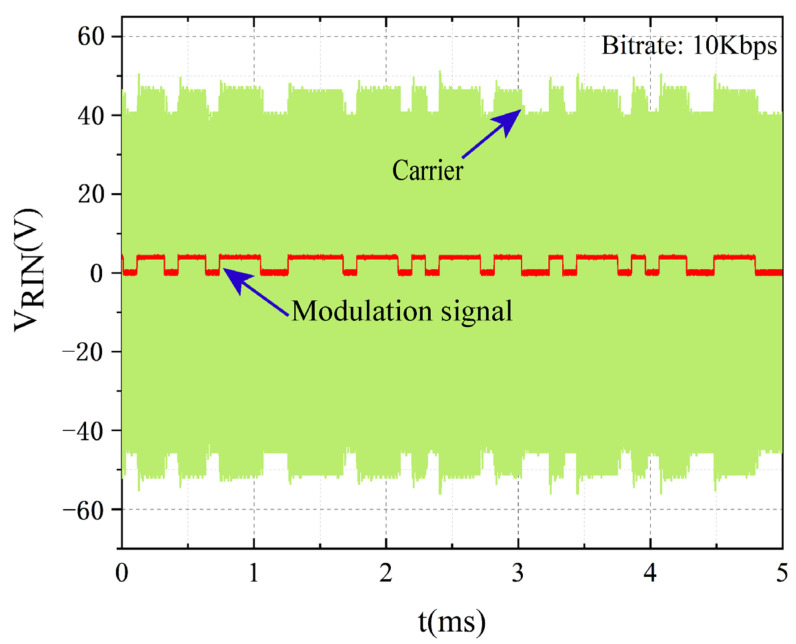
Carrier envelope at impedance modulation rate of 10 Kbps.

**Table 1 sensors-22-00727-t001:** Properties of acoustic-electrical channel components.

Component	Attributes
PZT	E-PZT material: PZT-4; 5-PZT material: PZT-5; Resonant frequency: 1 MHz; Diameter*thickness: 50 mm × 2.8 mm;
Metal wall	Material: 304 stainless steel; Length*Width*Thicness:150 mm × 200 mm × 11 mm;
Coupling agent	Brand: Araldite; Weight Ratio: ARALDITE^®^AV138M-1:HARDEBERHV998 = 2.5:1

**Table 2 sensors-22-00727-t002:** Mutual conversion relationship between *Z* parameters and *T* parameters.

	*Z* Parameter	*T* Parameter
** *Z* ** **parameter**	$\left[\begin{array}{cc} Z_{11} & Z_{12}\\ Z_{21} & Z_{22} \end{array}\right]$	$\left[\begin{array}{cc} \frac{A}{C} & \frac{AD-BC}{C}\\ \frac{1}{C} & \frac{D}{C} \end{array}\right]$
** *T* ** **parameter**	$\left[\begin{array}{cc} \frac{Z_{11}}{Z_{21}} & \frac{Z_{11}Z_{22}-Z_{12}Z_{21}}{Z_{21}}\\ \frac{1}{Z_{21}} & \frac{Z_{22}}{Z_{21}} \end{array}\right]$	$\left[\begin{array}{cc} A & B\\ C & D \end{array}\right]$

## Data Availability

The data presented in this study are available from corresponding authors upon reasonable request.

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
