# Peer review of "An Optimal Design Method for Improving the Efficiency of Ultrasonic Wireless Power Transmission during Communication"

_sensors, 2022, doi:10.3390/s22030727_

Round 1
Reviewer 1 Report
The authors present modelling and experimental results demonstrating the optimization of simultaneous power transfer and communication across a metallic barrier using ultrasonic waves. The concept is of wide interest and the paper is very well written, with the figures and data being presented in a clear and effective manner. I suggest the authors to consider the following recommendations before publication:
- The authors motivate the topic of ultrasonic wpt by mentioning prior work related to powering across metallic barriers. However, US-WPT has also been widely used for powering implantable medical devices and consumer electronics. It would improve the introduction if the authors make a mention of the other applications where US-WPT is widely used. For example, the authors could refer to the following articles:
Charthad, Jayant, et al. "A mm-sized implantable medical device (IMD) with ultrasonic power transfer and a hybrid bi-directional data link." IEEE Journal of solid-state circuits 50.8 (2015): 1741-1753.
Rekhi, Angad S., Butrus T. Khuri-Yakub, and Amin Arbabian. "Wireless power transfer to millimeter-sized nodes using airborne ultrasound." IEEE transactions on ultrasonics, ferroelectrics, and frequency control 64.10 (2017): 1526-1541.
Surappa, Sushruta, and F. Levent Degertekin. "Characterization of a parametric resonance based capacitive ultrasonic transducer in air for acoustic power transfer and sensing." Sensors and Actuators A: Physical 303 (2020): 111863.
- Authors present previous work dedicated to US communication across barriers- some of these methods make use of frequency modulation (FM) whereas other methods make use of amplitude modulation (AM). Authors themselves make use of AM based communication in this paper. It would be great for the reader if there was more information in the introduction related to the relative strength and weaknesses of these two methods with respect to power consumption, data transfer rate, circuit complexity etc. Talk about advantages/disadvantages of AM vs FM based methods
- Line 109 – ‘deteriorate’ should be ‘deteriorating’
- Similar to figure 5b, can the authors include a plot showing the experimentally measured efficiency and Ma for different RL?
Reviewer 2 Report
The paper presents a design method for improving the efficiency of U-WPT during communication by adopting the SCIM technology. The reviewer will provide some comments for the improvement of the current paper.
- In the paper, the SCIM technology in [14] and [34] is used for improving efficiency during communication. However, it is difficult to find further contribution and novelty. The authors should provide the contribution and novelty of the paper. Also, effectiveness of the proposed methods should be verified more clearly by using proposer analysis method and measurement.
- In order to explain Figure (5b), parameters which you used should be provided. Also, the authors should explain the downward/upward modulation method and how to get efficiencies from line 344 to line 362.
- The matching networks designed in Fig. 2(b) is different form them in Fig. 5(a). Can the authors explain why two networks are different?
- Is Equation 9 forward transmission coefficient in the case including the matching network? Is Equation (20) (The equation number should be modified!) necessary?
- Equation (2) and Table 2 would be replaced by adding references.
- Equations (3) and (5) are important. The authors should explain more in the paper how equations (3) and (5) are derived.
- Regarding the sentence from line 259 to line 260. There is no RL in equation (13). The authors should explain “… to change RL to change the channel input impedance.”.
- Is RIN in equation (11) real? If not, Zin is recommended?
- It is necessary to give explanation about the power amplifier which you used in experiment.
- In Table 1, Mass Ratio: ??
- Figure 1 and Figure 2 are similar. Figure 2 seems to provide less information. Also, an equivalent circuit of the system is necessary. It is recommended that Figure 2 should be changed into an equivalent circuit of the system in Figure 1.
- Also, in Figure 1, a function generator and a RF PA are separated, but in Figure 2, RF PA consists of an oscillator and a RF PA. Consider using a power source with Z0=50 ohm.
- Check units. For example, Kbps in line 20 and kbps in line 75
- Check: trans-mission in line 45, etc.
Round 2
Reviewer 2 Report
The reviewer agrees with the effort on the improvement of the paper. However, there are still some questions and comments.
- Do you have experiment results on efficiency according to RL as shown in Fig. 5 (b). In Fig. 8(b), only modulation coefficient was shown, not efficiency. The main idea of the paper is that using SCIM, both modulation coefficient and WPT efficiency will be improved.
- In Fig. 1, a function generator will have usually 50 ohm characteristic impedance. In Fig. 2, there is no characteristic impedance of the function generator. Rs is for power amplifier. In Fig. 4(a), RF PA is used. It seems to be confused. Consider RF power source instead of RF PA.
- Check the parament RLS on line 444.
- Can the authors mark the inductors and capacitors in Fig. 3(c) used in Fig. 2(b)?
- It seems that theory is a little different from experiment for RL>250 ohm in Fig. 8(b). The authors should comment on the difference.
- The authors should explain how to obtain 51.53 W and 56.25 % from line 457 to 458. What is input power? 37.6W and 45.75%?
- Check equations (12), (21), and (22)
Round 3
Reviewer 2 Report
The reviewer would like to say that the authors should read the paper carefully before publication. I think there seem to several mistakes which I cannot find. For example, see Eq.(22). I guess that it is a mistake (Efficiency is always more than 100%).
-. Eq. (22) should be checked.
-. Comparing L and C in Fig. 3(a) and Fig. 3(b), Ls and Cs are different. I assume that optimum frequency and peak efficiency will be changed. It is recommended that the simulation results with exact inductors and capacitors which you used should be added.
-. 91.6 W input power will be shown as the inset figure for output power in Fig. 8(a) is shown.
